# An Optimization Tool for Production Planning: A Case Study in a Textile Industry

**Rodrigo Ferro [1,\*], Gabrielly A. Cordeiro [1] , Robert E. C. Ordóñez [1,\*], Ghassan Beydoun [2] and Nagesh Shukla [2]**

[1] Department of Manufacturing Engineering and Materials, University of Campinas, Campinas 13083-860, Brazil; gcordeiro@fem.unicamp.br

[2] School of Information, Systems and Modelling, University of Technology Sydney, Sydney 123, Australia; ghassan.beydoun@uts.edu.au (G.B.); nagesh.shukla@uts.edu.au (N.S.)

\* Correspondence: r075450@dac.unicamp.br (R.F.); cooper@fem.unicamp.br (R.E.C.O.); Tel.: +55-19-3521-3492 (R.E.C.O.)

**Abstract:** The textile industry is an important sector of the Brazilian economy, being considered the fifth largest textile industry in the world. To support further growth and development in this sector, this document proposes a process for production analysis through the use of Discrete Event Simulation (DES) and optimization through genetic algorithms. The focus is on production planning for weaving processes and optimization to help make decisions about batch sizing and production scheduling activities. In addition, the correlations between some current technological trends and their implications for the textile industry are also highlighted. Another important contribution of this study is to detail the use of the commercial software Tecnomatix Plant Simulation 13®, to simulate and optimize a production problem by applying genetic algorithms with real production data.

**Keywords:** Discrete Event Simulation; genetic algorithm; production planning; digitalization

## 1. Introduction

To maintain market share, sustaining competitiveness is a key strategy of any industry. In the textile industry, sustaining competitiveness is challenging as the market is constantly shifting due to rapid changes in customer preferences, seasonality and fashion trends. Companies constantly seek to adopt the best practices to shorten time to market (TTM) of its products [1].

Brazil is the fifth largest textile and garment producer in the world. The sector produces 9.8 billion garments and is valued at US $40 billion per year. The industry is also the second largest employer in Brazil [2] and represents 16.7% of jobs and 5.7% of the Transformation Industry's turnover. It is estimated that this sector has a 2.5% share of GDP.

Brazil is a world reference in beachwear, jeanswear and homewear design, having also grown in the fitness and lingerie segments. Brazil is the largest complete Textile Chain in the West, from the production of fibers, such as cotton plantation, to fashion shows, passing through spinning mills, weaving mills, processing companies, clothing and large retail.

Regarding the current state of the textile industry in Brazil, there are no specific data that allow for a diagnosis or a comparison with other sectors. However, ABIT (Brazilian Association of Textile and Confection) has identified that, like the entire market, the textile sector has also focused on Industry 4.0, combining automation, data management and standardization of intelligent processes.

A prospective study carried out in 2016 shows that some institutions (ABIT, ABDI, SENAI, CETIQT) are promoting studies for the textile sector, but there is no real example, only application prospection, because this type of industry is also characterized with a low level of technological application [3]. The authors emphasize that technologies can increase customization and reduce inventories, which is in line with the Fast Fashion movement.

Through interviews, they detected that the structuring of local productive arrangements can be a good strategy for the development of digitalization in this sector.

Indeed, the importance of the textile industry in Brazil warrants further studies to improve its global competitiveness.

To improve the decision-making process and enhance agility of production as demand changes, computational tools can be used. Specifically, modeling and simulation tools can help to create a virtual representation of the production processes so that different scenarios can be analyzed and tested before they are deployed. Sakurada and Miyake (2009) emphasize the significant increase in the use of the simulation due to its contribution for decision making. Moreover, simulation and modeling have been highlighted as a key driver of Industry 4.0 due to its capacity to manage complex manufacturing processes and detect errors before they propagate in the manufacturing process [3,4].

This work applies Discrete Event Simulation (DES) software to optimize the production planning of weaving process. The proposed methodology includes DES together with an optimization procedure to evaluate and identify best scenario to support optimal decision making in a given manufacturing process. A simulation model is developed to optimize production scheduling and to reduce overtime during periods of high demand.

DES has long been considered as a simulation tool which can analyze complex process-related behaviors to identify process issues (bottlenecks, redundancies, inefficiencies) in the process-oriented systems. They have been successfully applied in a variety of manufacturing environments including scheduling policies and work order release [5], WIP and throughput-based performance evaluation of production systems, just-in-time (JIT) production system and capacity requirements analysis [6].

The proposed research methodology is underpinned by a case study which presents an analysis of the real context of textile industry using Tecnomatix Plant Simulation 13® by Siemens. The selection of this case study is an important contribution of the work as it details a practical application of the underlying theories used. The study also explores a specific functionality of the simulation software used: genetic algorithms.

In this context, this work shows an alternative for the use of computational tools to model, simulate and optimize the production planning process, since the biggest problem facing the company is that of obtaining a more assertive production planning to meet the demand in the long term, since currently, the production planning process is based on the empirical experience of the employees.

## 2. Background

### 2.1. Simulation and Optimization in Textile Industry

Simulation can be briefly defined as a computer-based simulation model of a real system which helps to analyze a given problem to support the risk-free decision making. Simulation modeling allows studying real systems and experimenting with it using computational models without the need for costly practical interventions [4,7–9] and is being recognized as methodology to find solutions of real problems from many sectors including transport, healthcare, manufacturing, supply chains, energy systems and sustainability [10].

There are different abstraction levels of simulation models and they depend on the detailing of the built model [11]. The higher the level of approximation between the simulation model and the real model, the greater the complexity and necessity for detail.

In this context, the Discrete Event Simulation (DES) is applied for scenarios of low or medium abstraction level. DES is used in systems in which event or state changes happen in a discrete way during the time, so no change occurs between two consecutive states [12], and the simulation which has been widely used to analyze activities of planning, implementation and operation with the highest application are manufacturing and logistics systems [13–16].

Commercially available DES software tools are commonly used in manufacturing environments as a simulation tool for decision making, with most of the commercial

software in the area of manufacturing engineering using DES. In the literature, there are some studies that compare the different DES software options [17,18].

Among these, the Tecnomatix Plant Simulation 13® by Siemens has good indicators in visual aspects and software compatibility, highlighting it by having a good graphical interface and ability to integrate with other software [17].

Fabric production scheduling based on computational simulation tools to solve the problem of finding a workable schedule to allocate different fabric orders to meet order due dates and maximize loom utilization in the textile industry has been a challenge since at least the end of the last century and some solutions have been proposed [19–21].

The application of DES to simulate the balance monitoring of the garment production line has been reported by some authors, making a comparison of production times monitored through manual time taking and through the sensing that was installed associated with the simulation software [22]. Some others have demonstrated the use of simulation for balancing the garment production line where the time of production processes are collected through RFID, in order to perform the simulation for five different scenarios for the same production line, making the redistribution between the stages of the operation [23].

More recently, the search for solutions has led to the use of more sophisticated tools, such as an improved ant colony algorithm or the application of flexible job shop modeling on scheduling a woven labeling process [24,25].

Optimization is one most of important aspects of simulation. The main objective of optimization is to find the optimal solution based on single or multiple objective functions given the input data, constraints and resources available.

Optimization methods can be classified into mathematical programming methods such as (Linear Programming—LP, Nonlinear Programming—NLP, Dynamic Programming—DP) and evolutionary or heuristic methods such as genetic algorithms and simulated annealing, among others [26].

The production planning and scheduling problems can also be modeled as an optimization problem with different level of constraints [27]. There are commercial software/packages available to optimize problems using robust optimization methods such as genetic algorithm (GA).

Due to difficulties in mathematically modeling the problem as well as computational time required, evolutionary computing-based methods such as GA are preferred over traditional mathematical programming approaches. Evolutionary methods such as GA can solve optimization problems in non-linear systems, non-differentiable or even discontinuous functions [28].

GA use mechanisms inspired by biological evolution, reproduction, mutation, recombination and selection. Their application has comprehensively reduced the computational time taken to resolve NP-hard problems while maintaining the quality of solutions obtained [29]. GA requires (i) a method to measure the quality of a potential solution, (ii) combination of solutions to generate new individuals in the population and (iii) selection criteria for retaining or removing solutions in the search process.

The difference of the solution studied in this study is in the fact of using commercial software that applies DES and genetic algorithms as an alternative to continue evolving in the search for solutions to solve the problem of weaving scheduling.

## 2.2. Industry Digitalization

Digitalization began in the 1970s with the introduction of controls and microprocessors in the industry, along with the evolution of IT [30]. The application of Information Technology (IT) in the production systems is one of the main aspects that contributes to digitalization of industry. Nowadays, digitalization is broader, covering activities of operations and product design, besides the relationship with the supply chain and customers [31]. In this context, one of the main challenges of digitalization is to cover the whole supply chain [30,31].

Basically, the digitalization consists in creation of a twin computational model to the physical environment [32]. This computational environment will be used to simulate and optimize scenarios without intervention in the real environment, thus allowing the extrapolation of the experiments without any consequences for the production system. Therefore, the digital transformation needs to be based on the following pillars: Capacity to collect, manage and analyze digital data; ability to work autonomously and in an organized way; connectivity and synchronization with the supply chain, and digital access to customers for more transparency and new products.

The complexity of digital industry generates a need for the use of intelligent environments, combining physical and cybernetic technologies [33,34]. Another important factor for the proper functioning of the digital industry is a good architecture in mining and data storage in the Big Data concept [35]. This data collection must be carried out in real time and, for this, stand out the MES and RFID tools [32,36].

Digitalization concepts and real time exchange data was applied in experimental environment [37]. It is estimated that over the next five years, the digitalization will provide an annual revenue of $493 billion and a reduction of $421 billion in operating costs. The key to this result is given by two factors: The digital transformation of the industry both in the factory environment and in the relationship with the customer through the products and knowhow protection, that is, the protection of the intellectual capacity of the company [36].

This context of industrial digitalization is also known by the term Industry 4.0. In the case of development of the textile industry in the boundary of Industry 4.0, it can be highlighted by some technologies applications, as the use of simulation software to enhance the environment of data exchange and generate a virtual model, as well as through the use of Cyber-Physical Systems (CPS), a structure that establishes a smart factory.

A survey done in Germany indicated flexibility as one of the most significant parameters to future success of the textile industry. In addition, they highlight the relevancy of automated process for this industry segment that have short product lifecycles, demand fluctuations and tendencies of customization [38].

The following aspect is very important to study the implementations of Industry 4.0 architecture in the textile industry. Industry 4.0 has been defined as a production system composed by smart machineries, smart products, storage systems and facilities able to exchange and control information autonomously [39]. An architecture model was proposed for the deployment of CPS that is structured in 5 levels (I—Smart Connection Level, II—Data-to-Information Conversion Level, III—Cyber Level, IV—Cognition Level, V—Configuration Level) and it is named 5C architecture [40].

Considering 5C architecture, the simulation tools for production planning can be identified in the cognition level to provide data for collaborative diagnostics and decision making. This shows the relevance of the development of researches about simulation for optimization of production processes which is a department that has several restrictions throughout the process.

In the abovementioned sense, the use of technology to improve connectivity and the data path through DES can allow the transformation of most of the static activities to dynamic in the manufacturing environment through integration with manufacturing management tools (Enterprise Resource Planning—ERP, Core Manufacturing Simulation Data—CMSD, Manufacturing Execution System—MES, Core Manufacturing Simulation Data—CMSD, e-Kanban and Radio Frequency Identification—RFID) [41].

## 3. Case Study

This section presents the details of the application of proposed modeling approach in the case of a textile manufacturing company. We first introduced the simulation model for planning and development, presenting characteristics of the textile manufacturing company, objectives of the study, and results and discussion.

The simulation model developed for this case study is based on the steps proposed by [9], in which the main objective is to ensure that the simulation model has all the necessary characteristics to be used as a tool for decision making. Using this approach, the development of the model will be guided by the problem to be solved and the validations will guarantee the adhesion between the real case study and virtual model developed. To improve the management of the activities necessary for the development of the simulation model, the stages of the proposed activities were classified as per the requirement of this case study (see Table 1).

**Table 1.** Steps for simulation model development.

| Flow of Activities | Description of Activities |
| --- | --- |
| Problem formulation and study plan | Mapping of industry characteristics; definition and analysis of the central problem of the project; definition of desired results, and mapping of problem variables. |
| Data collection and model definition | Collection of processing data; collection of product characteristics; collection of product storage and distribution characteristics, and definition of the modeling logic. |
| Conceptual model validation | Conceptual validation face-to-face and conceptual validation by sensitivity analysis. |
| Model development and verification | Construction of the simulation model. |
| Model validation | Operational validation and definition of the confidence interval. |
| Design experiments and make production runs | Definition of optimization techniques; insert the data for the use of the Genetic algorithm tool, and run the model. |
| Analyze output data | Analysis of output data and choice of the best scenario. |
| Document, present and implement results | Observation and analysis of results implemented. |

### 3.1. Formulating the Problem and Study Plan

This work was carried out in a textile company that has a production capacity of 7500 kg of fabric per month. Their main clients are small and medium-size enterprises spread all over Brazil. The fabric produced is used in the production of school uniforms, clothing for sportswear and children's fashion.

The fiscal departments of this knitwear use the unit of measure of kilograms as a form of control, however, the manufacturing department uses number of coils as their measurement unit. Each coil has capacity for 16 kg of fabric.

Currently, the company's production process relies on the employees' experience in the scheduling process. This model worked for a period, however, with the growth in the number of demand requests, production scheduling must be replaced by more assertive evidence-based planning approaches.

The difficulty of having a long-term demand forecast is one of the main problems for the company's production scheduling. As the company serves multiple customers, the number of orders is at an all-time high. All orders must be manufactured and delivered in the following month, making it difficult to achieve long-term planning for the production. As such, there is a large variation in production volume among the months. This variation can be observed in Figure 1, which shows the monthly production volume in the period of one year.

## Manufacturing per month (kg)

Figure showing a bar chart of monthly production values: July 3800, August 5800, September 5600, October 3700, November 5500, December 4800, January 9000, February 9000, March 8500, April 3500, May 8000, June 5800. A red line marks 7500.

**Figure 1.** Variation in production volume among the months.

It is evident that the months of January, February, March and May exceeded the maximum capacity of production of the company (7500 kg). In order to match this demand, it was necessary to work overtime.

In this case, we have proposed the optimization of production scheduling through the simulation model. Its application in this case study is expected to reduce the amount of overtime worked in the months of highest demand. It is also possible to identify ways of increasing client orders especially for the months which are below the production capacity level.

### 3.2. Data Collection and Model Definition

In this step, the required data for the construction of simulation model was collected. Figure 2 illustrates the flow of materials throughout the manufacturing processes in the textile manufacturing company.

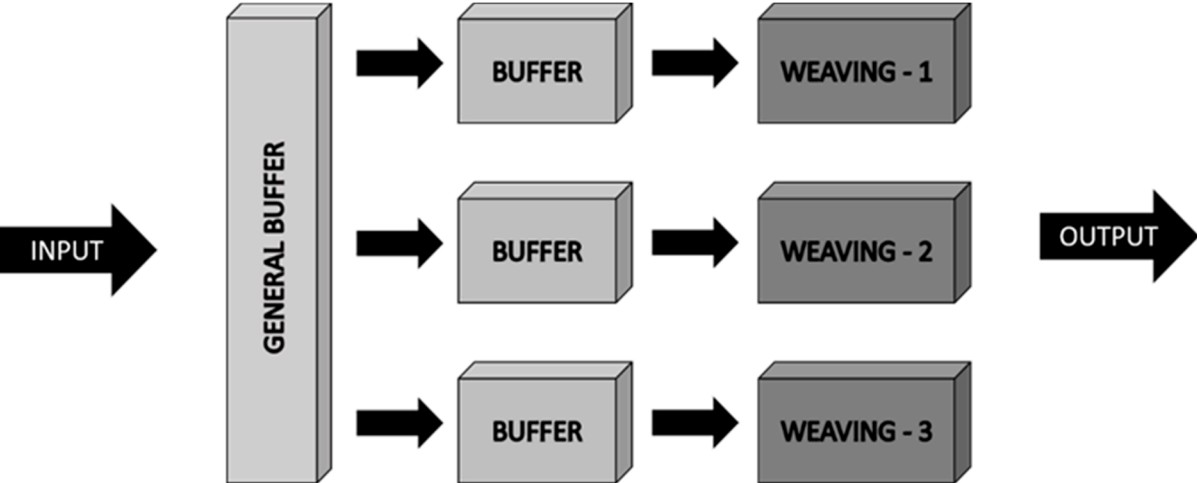

**Figure 2.** Flow of materials through the manufacturing processes.

First, monthly production data for one year stratified by weaving types was collected. Table 2 presents this production data in terms of kilograms of fabric produced. The company works in two shifts, with 80 h of work per week.

**Table 2.** Steps for simulation model development.

|  |  | Jul/16 | Aug/16 | Sep/16 | Oct/16 | Nov/16 | Dec/16 | Jan/17 | Feb/17 | Mar/17 | Apr/17 | May/17 | Jun/17 |
|---|---|---|---|---|---|---|---|---|---|---|---|---|---|
| Weaving 1 | PV | 1435 | 1802 | 2539 | 1833 | 2538 | 1311 | 4331 | 3843 | 3279 | 1593 | 3456 | 2332 |
|  | Cotton | 0 | 0 | 0 | 203 | 261 | 835 | 0 | 52 | 961 | 0 | 370 | 676 |
|  | Soft | 139 | 910 | 380 | 0 | 0 | 171 | 0 | 626 | 0 | 0 | 279 | 188 |
| Weaving 2 | PV | 661 | 1802 | 261 | 1488 | 427 | 842 | 2997 | 606 | 2954 | 80 | 1447 | 699 |
|  | Cotton | 0 | 773 | 0 | 0 | 830 | 0 | 0 | 209 | 0 | 0 | 701 | 952 |
|  | Soft | 0 | 0 | 546 | 0 | 616 | 243 | 0 | 1501 | 0 | 1023 | 386 | 501 |
|  | Piquet | 0 | 40 | 271 | 0 | 338 | 433 | 895 | 1958 | 0 | 507 | 326 | 150 |
|  | Cloth | 1417 | 285 | 1371 | 142 | 393 | 923 | 641 | 100 | 1303 | 291 | 703 | 255 |
| Weaving 3 | Rib_PV | 101 | 71 | 112 | 95 | 128 | 12 | 136 | 71 | 150 | 124 | 321 | 0 |
|  | Rib_Cot | 0 | 0 | 0 | 0 | 0 | 24 | 0 | 106 | 0 | 22 | 0 | 0 |

Then, the data for batch producing equipment was collected (see Table 3). In this case, the following data were used: Processing time of each type of product in different weavings; setup time among coils produced in the same batch; scrap percentage for batch production, and time of corrections and inspections on each coil.

**Table 3.** Weavings' data.

|  |  | Processing Time per Coil (Sec.) | Setup Time per Coil (Sec.) | Scrap Percentage (%) | Inspection Time per Coil (Sec.) |
|---|---|---|---|---|---|
| PV | Weaving 1 | 3330 | 120 | 3.1% | 60 |
|  | Weaving 2 | 3610 | 150 | 4.8% | 60 |
| Cotton | Weaving 1 | 4040 | 120 | 5.5% | 120 |
|  | Weaving 2 | 4140 | 180 | 5.2% | 60 |
| Soft | Weaving 1 | 2860 | 60 | 2.5% | 150 |
|  | Weaving 2 | 2610 | 60 | 3.0% | 150 |
| Piquet | Weaving 2 | 3360 | 150 | 6.1% | 60 |
| Cloth | Weaving 2 | 5300 | 60 | 7.2% | 240 |
| Rib_PV | Weaving 3 | 9300 | 300 | 3.2% | 600 |
| Rib_Cot | Weaving 3 | 9900 | 300 | 3.5% | 600 |

Table 4 shows the setup time used when there is a change of fabric type. Another relevant information for the construction of the model is the percentage of the availability of the machines related to the maintenance performed. These data are: Weaving 1–12%; Weaving 2–8%, and Weaving 3–15% of the total work time stopped for maintenance.

**Table 4.** Setup time for production (in minutes).

|  | PV | Cotton | Soft | Piquet | Cloth | Rib_PV | Rib_Cot | - |
|---|---|---|---|---|---|---|---|---|
| - | 20:00.0000 | 20:00.0000 | 20:00.0000 | 20:00.0000 | 20:00.0000 | 20:00.0000 | 20:00.0000 | 0.0000 |
| PV | 0.0000 | 1:00:00.0000 | 1:00:00.0000 | 2:00:00.0000 | 3:00:00.0000 | 0.0000 | 0.0000 | 5:56:00.0000 |
| Cotton | 1:00:00.0000 | 0.0000 | 1:00:00.0000 | 2:00:00.0000 | 3:00:00.0000 | 0.0000 | 0.0000 | 4:00:00.0000 |
| Soft | 1:00:00.0000 | 1:00:00.0000 | 0.0000 | 2:00:00.0000 | 3:00:00.0000 | 0.0000 | 0.0000 | 0.0000 |
| Piquet | 2:00:00.0000 | 2:00:00.0000 | 2:00:00.0000 | 0.0000 | 3:00:00.0000 | 0.0000 | 0.0000 | 3:00:00.0000 |
| Cloth | 3:00:00.0000 | 3:00:00.0000 | 3:00:00.0000 | 3:00:00.0000 | 0.0000 | 0.0000 | 0.0000 | 0.0000 |
| Rib_PV | 0.0000 | 0.0000 | 0.0000 | 0.0000 | 0.0000 | 0.0000 | 1:00:00.0000 | 50:00.0000 |
| Rib_Cot | 0.0000 | 0.0000 | 0.0000 | 0.0000 | 0.0000 | 1:00:00.0000 | 0.0000 | 25:00.0000 |

### 3.3. Development of the Simulation Model

After the data collection of the production system, next step was the development of the simulation model. For this, we modeled the flow of materials observed in the company.

Figure 3 presents the simulation model developed. In this model, it is evident that the parts enter into the production process at the source based on a schedule defined in the table list (Table_List) indicated with the letter "D" in Figure 3. This table constitutes the information about the production order for each of the month under analysis.

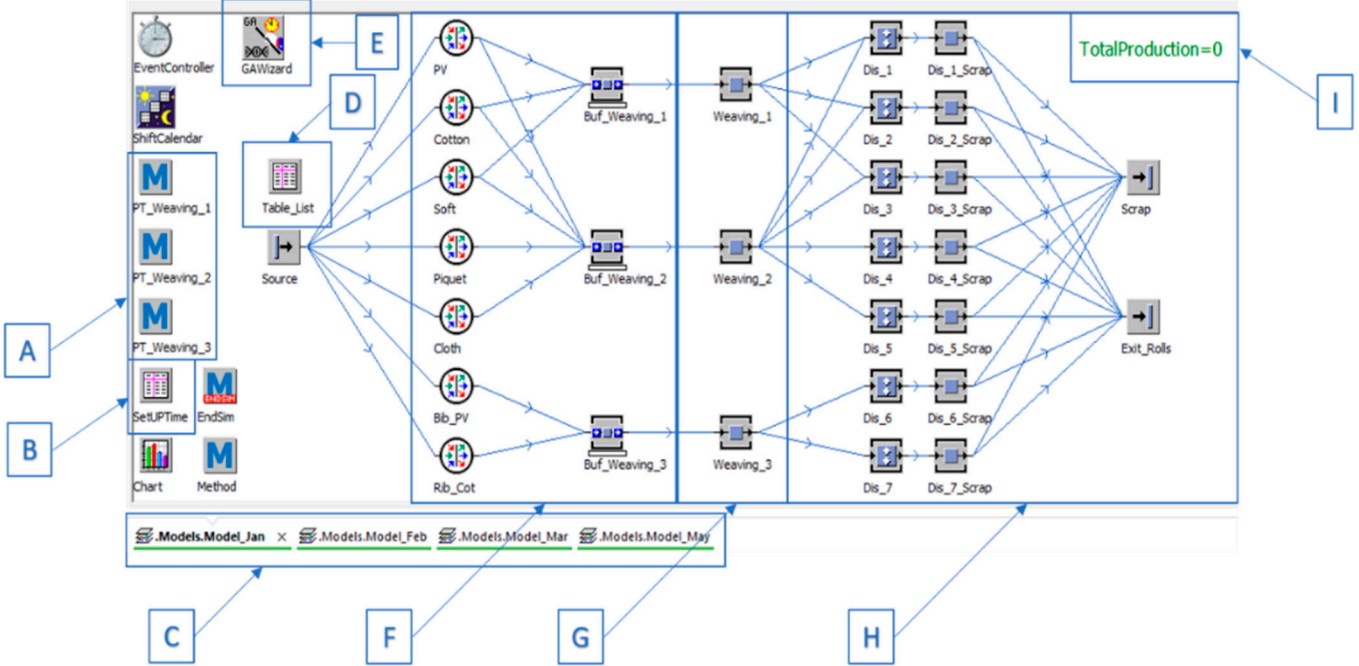

**Figure 3.** Simulation model.

The pieces that enter the system are divided according to the type of fabric; this happens in the "F" section of the model. We have introduced some buffer in "F" section so that these pieces wait until the next processing machine is available. After this step, the material can be manufactured in the weavings which is shown in region "G". The operating parameters of the weavings vary according to the type of fabric to be manufactured. Therefore, the setup time parameter is controlled by the SetUpTime table list (see Table 4) and in the "B" area of the model. For the control of the process time, the control methods were developed which are shown in region "A" of the model, shown in Figure 4.

After the fabric is made, the coils are sent to the "H" region in which the production is separated according to the type of product, in addition to separating the scraps. To control the amount of fabric produced, TotalProduction can be seen in the region "I" of the model.

The region "C" provides different frames of the model, which represents the simulated months (January, February, March and May). The simulation was conducted for the months where demand exceeded the production capacity of the company resulting in overtime.

The GAWizard, which can be seen in the region "E" of the simulation model, is the tool that uses GA for the optimization of production sequencing. In order to run GA, it is necessary to define/select system variables which are to be optimized. We have selected overall manufacturing time taken to fulfil the production order as the main objective for optimization with GA. The sequence of production orders was adopted as the optimization variable. That is, by varying the order of production sequences we expect to find the best sequence, which optimizes the overall manufacturing time.

Figure 4 shows the variation of the processing time data used for weaving operation based on the type of fabric to be produced.

```
is                                is                                is
do                                do                                do

inspect @.name                    inspect @.name                    inspect @.name
  when "PV" then                    when "PV" then                    when "PV" then
      Weaving_1.ProcTime := 3330        Weaving_2.ProcTime := 3610        Weaving_3.ProcTime := 0
  when "Cotton" then                when "Cotton" then                when "Cotton" then
      Weaving_1.ProcTime := 4040        Weaving_2.ProcTime := 4140        Weaving_3.ProcTime := 0
  when "Soft" then                  when "Soft" then                  when "Soft" then
      Weaving_1.ProcTime := 2860        Weaving_2.ProcTime := 2610        Weaving_3.ProcTime := 0
  when "Piquet" then                when "Piquet" then                when "Piquet" then
      Weaving_1.ProcTime := 0          Weaving_2.ProcTime := 3360        Weaving_3.ProcTime := 0
  when "Choth" then                 when "Choth" then                 when "Choth" then
      Weaving_1.ProcTime := 0          Weaving_2.ProcTime := 5300        Weaving_3.ProcTime := 0
  when "Rib_PV" then                when "Rib_PV" then                when "Rib_PV" then
      Weaving_1.ProcTime := 0          Weaving_2.ProcTime := 0          Weaving_3.ProcTime := 9300
  else                              else                              else
      Weaving_1.ProcTime := 0          Weaving_2.ProcTime := 0          Weaving_3.ProcTime := 9900
  end;                              end;                              end;

end;                                                                end;
                                  end;
```

**Figure 4.** Programming code for processing time control of weavings.

### 3.4. Model Verification and Validation

To ensure that the simulation model can be used as a decision-making tool, it is important to know if the model results are valid with respect to the real production data. This means we have to compare model results with that of production data not used for model development.

The literature presents model verification and validation as the two approaches to understand whether the simulated model is properly developed given the conceptual model, and then, whether the model results are valid enough to make reasonable predictions.

Verification aims to verify if the modeling logic is correct and validation was used to define the confidence interval of the model. Both are described below:

Verification by graphic animation: This type of verification ensures that the simulation model has the same logic of movement of the real system. Using Plant Simulation model development environment, all decisions and movements of pieces were analyzed in low animation speed, as they travelled through the production environment.

Confidence level: At this time, the simulation model was validated for a 95% confidence level. For this purpose, the software tool "Experiment Manager" was used in which 20 experiments were conducted for each month. All the data collected during this exercise is shown in Table 5.

It is important to note that all values used in Table 5 are within the stipulated confidence level, because the software restricts values that are outside this range.

**Table 5.** Simulated production data for months analyzed.

|  | January | February | March | May |
|---|---|---|---|---|
| Exp.1 | 8163 | 7428 | 8196 | 7279 |
| Exp.2 | 8509 | 7972 | 8377 | 7296 |
| Exp.3 | 8476 | 7873 | 8410 | 7411 |
| Exp.4 | 8509 | 7971 | 8361 | 7262 |
| Exp.5 | 8575 | 8134 | 8361 | 7345 |
| Exp.6 | 8493 | 8166 | 8361 | 7295 |
| Exp.7 | 8526 | 8117 | 8410 | 7443 |
| Exp.8 | 8690 | 8134 | 8312 | 7377 |
| Exp.9 | 8410 | 8003 | 8427 | 7328 |
| Exp.10 | 8575 | 8166 | 8443 | 7460 |
| Exp.11 | 8295 | 7873 | 8410 | 7394 |
| Exp.12 | 8378 | 7857 | 8295 | 7246 |
| Exp.13 | 8427 | 8021 | 8377 | 7394 |
| Exp.14 | 8674 | 8263 | 8427 | 7476 |
| Exp.15 | 8427 | 8198 | 8410 | 7328 |
| Exp.16 | 8592 | 8086 | 8443 | 7493 |
| Exp.17 | 8542 | 8118 | 8460 | 7476 |
| Exp.18 | 8509 | 7971 | 8460 | 7427 |
| Exp.19 | 8509 | 8118 | 8328 | 7229 |
| Exp.20 | 8394 | 8182 | 8410 | 7329 |
| Average | 8483.65 | 8032.55 | 8383.9 | 7364.4 |
| Standard Deviation | 122.39 | 183.97 | 64.42 | 82.29 |
| Minimum Value | 8163 | 7428 | 8196 | 7229 |
| Maximum Value | 8690 | 8263 | 8460 | 7493 |

### 3.5. Results and Discussion

After the model validation, the next step was to use the optimization tool to find the best production sequence for the production operation. The GAWizard tool of simulation software by SIEMENS Plant Simulation® was used. It follows the steps shown in Figure 5. Some of the parameters involved in the process were: (i) one hundred generations; (ii) population size of ten individuals; (iii) elitist system was used for selection, where the best solutions are used to generate offspring for the next generation; (iv) single objective optimization, and (v) the best solution is stored which maximizes the throughput.

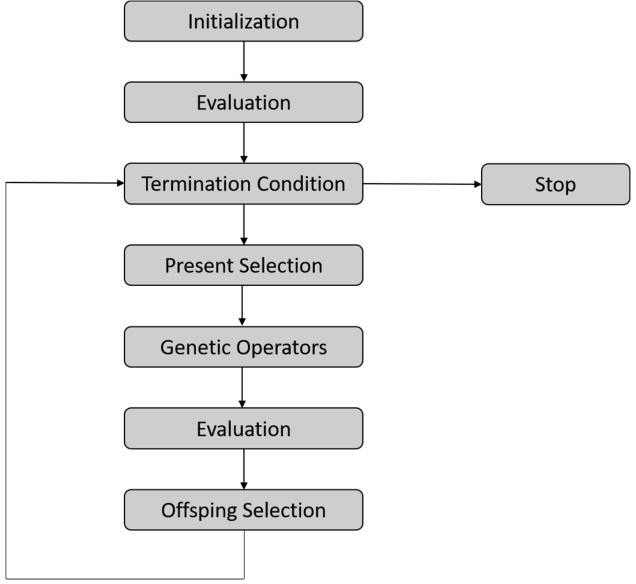

**Figure 5.** Steps for genetic algorithm—GAWizard (SIEMENS, 2011).

In Figure 6, we can observe the evolution of optimized solutions against the number of generations. It is evident that one hundred generations are enough for obtaining the optimization convergence.

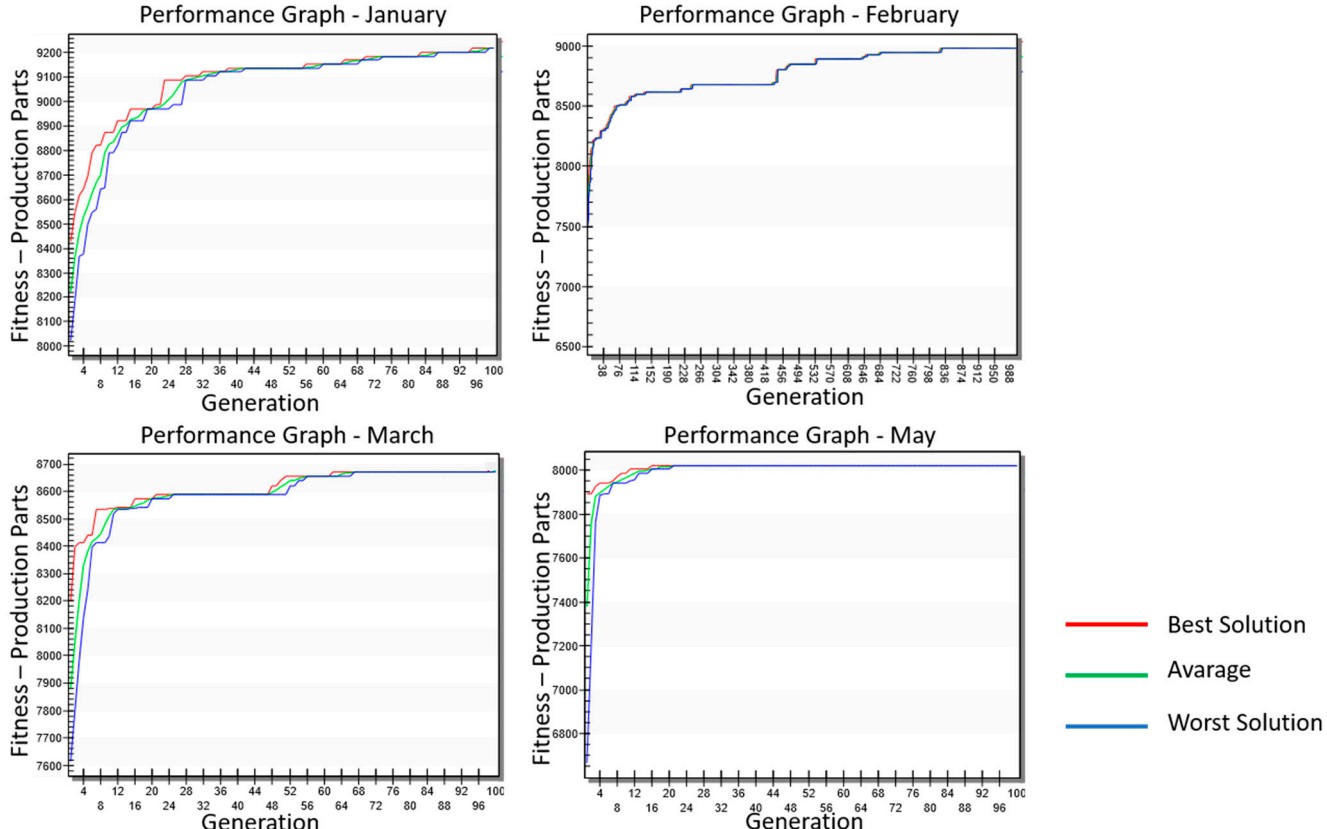

**Figure 6.** Evolution of the results in the months analyzed.

Moreover, as generations advance, the difference between worst solution and best solution decreases; in other words, the individual's fitness looks similar.

Table 6 presents the results of optimization for January. It shows that production mix and lot sizes used initially by the company compared to the optimized production mix found after the optimization procedure. The optimized production mix/sequence takes the least amount of time for monthly production.

**Table 6.** Optimization results for January.

| JANUARY | | | |
|---|---|---|---|
| **Start Production MIX** | **Number** | **Optimized Production MIX** | **Number** |
| .MUs.PV | 10 | .MUs.PV | 10 |
| .MUs.PV | 10 | .MUs.PV | 10 |
| .MUs.Piquet | 5 | .MUs.PV | 10 |
| .MUs.Cloth | 5 | .MUs.PV | 10 |
| .MUs.PV | 10 | .MUs.PV | 10 |
| .MUs.PV | 10 | .MUs.PV | 10 |
| .MUs.PV | 10 | .MUs.PV | 10 |
| .MUs.Rib_PV | 3 | .MUs.PV | 10 |
| .MUs.Piquet | 5 | .MUs. Piquet | 10 |
| .MUs.Piquet | 5 | .MUs.Piquet | 10 |
| .MUs.Piquet | 5 | .MUs.Cloth | 5 |
| .MUs.PV | 10 | .MUs.Cloth | 5 |

With the results of the optimization, Table 7 was obtained presenting the production performance before and after optimization. It is observed that the amount of fabric manufactured is higher and the amount of overtime worked is lower after the optimization. For example, in February, it took the company an extra 91 h to complete the planned production, whereas after the optimization, it took 15 h. Similar improvements were also observed in the other three months simulated.

**Table 7.** Data after optimization.

| | Planned Production Qty (kg) | Before Optimization | | After Optimization | |
|---|---|---|---|---|---|
| | | Production in Scheduled Working (kg) | Overtime (Hours) | Production in Scheduled Working (kg) | Overtime (Hours) |
| January | 9000 | 8163 | 77 | 9218 | 0 |
| February | 9000 | 7428 | 91 | 8749 | 15 |
| March | 8500 | 8196 | 14 | 8686 | 0 |
| May | 8000 | 7279 | 21 | 8019 | 0 |

Figure 7 shows the evolution of results with respect to a varying number of generations. For January, the production system produces 8163 units without the application of any optimization (GA) procedure. When GA was applied, the same production system could produce 8971 units in 20 GA generations, and 9271 units in 100 generations. In February, the optimization process took 826 generations to converge at its optimum best. However, for March and May, the optimized solutions were obtained in generations 68 and 16, respectively.

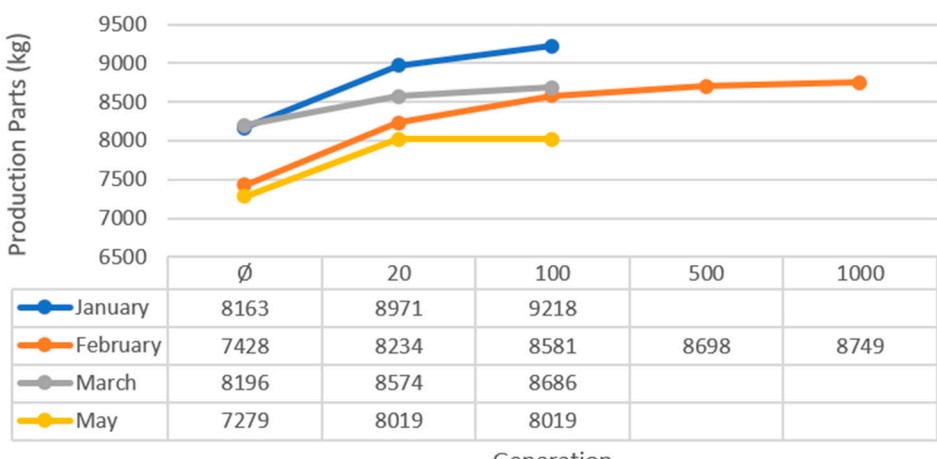

**Figure 7.** Evolution of the results by different GA generations.

It was also identified that even if the number of units produced increases with increased number of generations, it is of no value as the quantity of units produced will become greater than the quantity ordered by the customers.

Based on the case study presented in this work, it can be observed how the modeling and simulation process plus the optimization process can contribute to improve efficiency in long-term production scheduling.

We chose a traditional sequence of steps to create the simulation model representing the material flow and we used commercial software to verify and validate that model through the simulation environment and a tool to validate the level of confidence provided by the same software. Real data of the variation of the production volume was used and an optimization technique was applied through genetic algorithms (GA WIZARD) which is included in this software. The steps to use this technique were also detailed.

Thus, the company was able to increase its productivity, remembering that this company had a production limit of around 7500 kg of fabric per month; this limit occurs when using only one work shift. In months where demand exceeded this limit, overtime was required to meet demand. After the application of the optimization, a new sequence was generated, thus, the waste of time was reduced, consequently, less overtime was needed to meet the demand in the critical months.

## 4. Conclusions

This paper presented an interesting application of the commercially available software platforms for simulation-optimization of the textile manufacturing process. The study operationalized a bespoke simulation model to reduce production time. It effectively used GA for optimizing the production sequence/mix which can reduce the overall production time. It evaluated various alternatives associated with the model variables and finally the best alternative for production mix was proposed.

The results showed significant cost savings in the high season, in the months of January, February, March and May. The optimization process led to a reduction of overtime of 0, 15, 0 and 0 h, respectively. The study focused on the high demand period where cost savings are most needed. Other months were not addressed in this study because there were no production bottlenecks reported. This work confirmed that the proposed simulation model can contribute to the reduction of production time when it is used as a support tool for the production sequencing.

This study does not intend to compare methods for programming or optimization of production scheduling; it is intended to show the steps to use a computational tool for simulation and optimization of production scheduling, as well as the data chosen to facilitate a replication of the procedure by part of potential researchers or business users so that the data generated by the analysis can be used to connect to other interfaces or computational platforms in the context of digital transformation.

**Author Contributions:** Conceptualization, R.F. and R.E.C.O.; methodology, R.F. and G.A.C.; software, R.F.; validation, G.B. and N.S.; formal analysis, G.B. and N.S.; investigation, R.F., G.A.C. and R.E.C.O.; resources, R.F., R.E.C.O. and G.B.; data curation, R.F. and R.E.C.O.; writing—original draft preparation, R.F., N.S. and G.A.C.; writing—review and editing, R.F., G.A.C. and R.E.C.O.; project administration, R.E.C.O.; funding acquisition, G.B., R.E.C.O. and R.F. All authors have read and agreed to the published version of the manuscript.

**Funding:** This research was funded in part by São Paulo Research Foundation (FAPESP), grant number 2019/10088-9.

**Institutional Review Board Statement:** Not applicable.

**Informed Consent Statement:** Not applicable.

**Data Availability Statement:** Data sharing not applicable.

**Conflicts of Interest:** The authors declare no conflict of interest.

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
