# Peer review of "An Optimization Tool for Production Planning: A Case Study in a Textile Industry"

_applsci, doi:10.3390/app11188312_

Round 1

Reviewer 1 Report

This paper presents a simulation and optimization methodology to optimize production planning in a weaving factory.

The objective, while not innovative, is interesting and is worth investigating. Although, several weaknesses should be resolved before publishing. First, in the abstract, it should be clarified that GAWizard or at least genetic algorithm will be used. Also, while the abstract and simulation specifies the simulation tool, there is no indication on how the simulation was built. Secondly, the planification problem is not clearly presented. Thirdly, literature review should present simulation papers related to weaving planning.

Finally, the English should be reviewed entirely since there are a lot of writing mistakes.

Reviewer 2 Report

This paper presented an interesting application of the commercially available software platforms for simulation-optimisation of the textile manufacturing process. The experimental results are also provided.

Key comments:

  1. The results have confirmed that the proposed simulation model can contribute to the reduction of production time
  2. The topic of this paper is quite interesting
  3. The authors have presented their method in a detailed format

Things can be improved:

  1. In Conclusion Section, "This work confirmed that the proposed simulation model can contributes to the reduction of production time", not "contributes", but "contribute"

Reviewer 3 Report

Please give a more detailed overview of the position of the textile companies in Brazil. If possible and available, refer to their level of digitisation in terms of Industry 4.0 and compare it with other industries. Chapters 2.1 and 2.2. aren't necessary, you should give a short literature review of the previous uses of the simulation in the textile industry for optimization and scheduling, as well as the algorithms used and by that you confirm the validity of your approach.
A detailed and more critical approach to results is needed in "Discussion" chapter. In the "Conclusion" chapter, the limitations of this work, as well as the possibilities of for the future work must be given. 
